# Structure of Lipopolysaccharide from *Liberibacter crescens* Is Low Molecular Weight and Offers Insight into *Candidatus* Liberibacter Biology

**DOI:** 10.3390/ijms222011240

**Published:** 2021-10-18

**Authors:** Ian M. Black, Christian Heiss, Mukesh Jain, Artur Muszyński, Russell W. Carlson, Dean W. Gabriel, Parastoo Azadi

**Affiliations:** 1Complex Carbohydrate Research Center, University of Georgia, Athens, GA 30602, USA; ianblack@uga.edu (I.M.B.); cheiss@ccrc.uga.edu (C.H.); muszynski@ccrc.uga.edu (A.M.); rcarlson@ccrc.uga.edu (R.W.C.); 2Department of Plant Pathology, University of Florida, Gainesville, FL 32611, USA; mjain@ufl.edu (M.J.); dgabr@ufl.edu (D.W.G.)

**Keywords:** lipopolysaccharide, lipid A, nuclear magnetic resonance, Gram-negative bacteria, bacterial pathogenesis

## Abstract

Huanglongbing (HLB) disease, also known as citrus greening disease, was first reported in the US in 2005. Since then, the disease has decimated the citrus industry in Florida, resulting in billions of dollars in crop losses and the destruction of thousands of acres of citrus groves. The causative agent of citrus greening disease is the phloem limited pathogen *Candidatus* Liberibacter asiaticus. As it has not been cultured, very little is known about the structural biology of the organism. *Liberibacter* are part of the *Rhizobiaceae* family, which includes nitrogen-fixing symbionts of legumes as well as the *Agrobacterium* plant pathogens. To better understand the *Liberibacter* genus, a closely related culturable bacterium (*Liberibacter crescens* or Lcr) has attracted attention as a model organism for structural and functional genomics of Liberibacters. Given that the structure of lipopolysaccharides (LPS) from Gram-negative bacteria plays a crucial role in mediating host-pathogen interactions, we sought to characterize the LPS from Lcr. We found that the major lipid A component of the LPS consisted of a pentaacylated molecule with a β-6-GlcN disaccharide backbone lacking phosphate. The polysaccharide portion of the LPS was unusual compared to previously described members of the *Rhizobiaceae* family in that it contained ribofuranosyl residues. The LPS structure presented here allows us to extrapolate known LPS structure/function relationships to members of the *Liberibacter* genus which cannot yet be cultured. It also offers insights into the biology of the organism and how they manage to effectively attack citrus trees.

## 1. Introduction

*Candidatus* Liberibacter spp. are Gram-negative, fastidious and uncultured α-Proteobacteria (order *Rhizobiales*) that have lately emerged as a versatile group of insect-transmitted and phloem-limited plant pathogens. Citrus huanglongbing (HLB) or “greening” is one of the most destructive citrus diseases worldwide and is associated with *Candidatus* Liberibacter asiaticus (CLas) in Asia and the Americas, Ca. L. americanus (CLam) in Brazil, and Ca. L. africanus (CLaf) in Africa [1]. CLas is vectored and transmitted among citrus and some other *Rutaceae* species by the Asian citrus psyllid (*Diaphorina citri* Kuwayama). Ca. L. solanacearum (CLso) has a wide host range and infects several economically important *Solanaceae* and *Apiaceae* crops causing zebra chip disease on potato, psyllid yellows in tomato, yellows decline in carrots, and vegetative disorders in fennel, celery, parsnip, and parsley [2].

Systemic colonization of phloem sieve tube elements in HLB-infected citrus trees results in aberrant assimilate partitioning and nutrient transport [3]. HLB symptoms typically include yellow shoot sectors, asymmetric blotchy mottling of leaves, vein corking, stunted growth, premature fruit drop, lopsided fruits with abortive seeds, and uneven coloration leading to progressive decline in productivity and eventual death of the infected tree. HLB was first detected in Florida in 2005, and since then has rapidly spread to other citrus-growing states in the United States. Florida citrus acreage has decreased by 40% and production by 49% since their historical peaks in the last 20 years due to widespread HLB infection and lack of an effective management strategy [4]. Despite extensive research efforts to find an effective foliar treatment for the disease [5,6,7], no cure has yet been reported. Prevention efforts have largely centered on vector control and the destruction of infected trees to limit disease spread.

The surface of Gram-negative bacteria, which includes CLas, consists of an asymmetric outer membrane in which the inner leaflet is primarily comprised of phospholipids and the outer leaflet of lipopolysaccharide (LPS) [8]. The asymmetric nature of the outer membrane plays an important role in decreasing the permeability of the outer membrane to hydrophobic toxins [9], and proper biosynthesis of LPS is essential for the viability of most Gram-negative bacteria [10]. Notably, genomic sequencing of CLam revealed the near-total absence of LPS-related genes [11], indicating CLam may lack LPS and must therefore compensate via some unknown structural modifications. The structure of the LPS molecules is typically divided into three regions; the membrane-embedded, hydrophobic Lipid A, the largely structurally conserved (among strains of a single species) core oligosaccharide region, and a variable (even among strains of a single species) polysaccharide region comprised of repeating oligosaccharide units [12]. Bacteria of the genus *Liberibacter* are members of the *Rhizobiaceae* family [13], and as such, the lipid A produced by these organisms likely contains some structural features not found in most other pathogenic bacteria. For example, the related bacteria *Rhizobium* sp. Sin–1 contains lipid A that lacks the typical phosphate moieties found at the reducing and non-reducing glucosamine residues that comprise the β-6-GlcN disaccharide head of the lipid A [14]. Other bacteria of the *Rhizobiaceae* family (*R. leguminosarum* biovar viciae, *R. etli,* etc.) can replace the missing phosphate with galacturonic acid residues. These bacteria can further modify their lipid A by oxidizing the proximal glucosamine to 2 aminogluconate [14]. The lipid A of *Rhizobiaceae* family bacteria also contains a single secondary very-long-chain fatty acid (VLCFA, e.g., 27-hydroxyoctacosanoic acid) [14], which itself can be modified by the addition of an ester-linked β-hydroxyl butyric acid. Finally, some members of *Rhizobiaceae* (*Mesorhizobium loti*) contain 2,3-diamino-2,3-dideoxy-D-glucose in place of the traditional glucosamine. The presence of the polysaccharide region of the LPS and some of the lipid A modifications, in particular the VLCFA, have been shown to play important roles in bacterial survival and symbiotic/pathogenic infection as modification can attenuate or eliminate the ability of the bacteria to properly colonize plants and fix nitrogen [14].

All known pathogenic *Ca.* Liberibacter spp. remain uncultured to date [15] and only low-titer, transient cocultures have been reported with CLas as a minor component of a complex microbial community [16,17]. Only a single isolate of a single species of *Liberibacter* genus, *Liberibacter crescens* (Lcr) strain BT-1, has been axenically cultured in vitro [18]. Although Lcr was originally isolated from the leaf sap of Babaco mountain papaya, it has no known insect or plant hosts. Lcr BT-1 is naturally competent for transformation and is well established as a surrogate host for pathogenic *Ca*. Liberibacter spp. [19]. Genomic analyses have revealed a characteristic tendency for continued reductive genome evolution within all sequenced strains of *Ca*. Liberibacter spp. (1.2 Mb) in favor of an intracellular lifestyle, although microsynteny and significant phylogenetic similarities with Lcr genome (1.4 Mb) are also discernible [20]. Structural integrity and modifications of the core LPS moiety in Gram-negative bacteria in general [21], and more specifically of *Rhizobiales* have been shown to have a significant bearing on bacterial survival in vitro and ability to successfully form intracellular pathogenic/endosymbiotic infections [22]. The primary objective of the current study was to elucidate the complete structure of LPS from Lcr in order to gain insight into the recalcitrance of *Ca*. Liberibacter spp. to axenic culturing. The purified LPS from Lcr was also used to study LPS-mediated immune signaling networks *in planta* and the data will be published separately elsewhere (Jain et al., submitted).

## 2. Results

### 2.1. Purified LPS from Lcr Is Low Molecular Weight

To get an estimation of the size of the LPS from Lcr, and specifically, whether it synthesizes a complete high molecular weight *O*-polysaccharide, or only a truncated low molecular weight LPS/LOS, we compared its DOC-PAGE profile (Figure 1, Lane 6) with PAGE profiles of well-characterized S- and R- type lipopolysaccharides. We used *Salmonella enterica* serovar Minnesota S-strain (Figure 1, Lanes 1 and 7); *E. coli* EH100 (Ra mutant) (Figure 1, Lane 2), *S. minnesota* R5 (Rc mutant) (Figure 1, Lane 3); *S. minnesota* R7 (Rd mutant) (Figure 1, Lane 4); and *S. enterica* Re 595 (Re mutant) (Figure 1, Lane 5) as standards. The comparative analysis demonstrated that most of the Lcr LPS resolved in two distinct LMW bands at the bottom of the gel. The upper LMW band ran similarly to the LMW band of the Ra mutant of *E. coli* EH100. This mutant produces LPS deprived of the O-chain, with a lipid A and full core oligosaccharide (approximate *M*_W_ of 3.9 kDa; compare Lane 6 with Lane 2). The location of the lower band was consistent with the LMW bands of the Rd-Re mutants of *S. minnesota* (approximate maximum *M*_W_ 3–3.165 kDa; compare Lane 6 with Lanes 4 and 5, respectively). In addition to the two LMW bands, we observed a very faint ladder pattern in the upper part of the PAGE (Figure 1, Lane 6) Figure 1, Panel B shows the same gel after overexposure to the silver stain. The overexposure allowed for better visualization of the higher molecular weight ladder pattern seen faintly in Panel A. This suggests the presence in trace amounts of LPS with a higher molecular weight *O*-polysaccharide or other contaminating polysaccharides.

To further determine the molecular weight of the Lcr LPS, we performed size exclusion chromatography (SEC) of the *O-*polysaccharides from Lcr and the *E. coli* EH100 Ra mutant after mild acid hydrolysis and removal of the lipid A. These results can be seen in Figure 2. As with the DOC-PAGE gel (Figure 1), the SEC analysis shows the two released polysaccharides from both species to be of similar molecular weight. The released polysaccharide from Lcr has a slightly larger range of molecular weights, with most of the sample coming out as a sharp, early eluting peak in the column void volume (between 6 and 8 min, Figure 2) and the rest of the released polysaccharide eluting between 8 and 12 min. The *E. coli* EH100 core has an early eluting peak that comes out slightly later than the void volume of the column and ends slightly earlier than the Lcr polysaccharide. This analysis is consistent with the DOC-PAGE gel showing the two main Lcr bands having a slightly larger range than the *E. coli* EH100 Ra mutant. The SEC analysis confirms the size of the Lcr oligosaccharide is small, with the LMW portion of the released *O*-polysaccharide smaller than the 3.9 kDa *E. coli* EH100 Ra mutant, but the HMW portion of the released *O*-polysaccharide larger than the void volume of the column.

Taken together, the results of the DOC-PAGE gel and the SEC analysis revealed that the molecular weight of the Lcr LPS is relatively small, with some portion of the LPS larger in range and molecular weight than that of 3.9 kDa molecular weight of the *E. coli* EH100 Ra mutant, and without significant amounts of larger molecular weight *O*-polysaccharide.

### 2.2. Composition Analysis of the Purified LPS from Lcr

The glycosyl and fatty acid composition analysis identified 3-deoxy-α-*d*-*manno*-oct-2-ulosonic acid (Kdo), *l*-*glycero*-α-*d*-*manno*-heptopyranose (Hep), *N*-acetylglucosamine (GlcNAc), as well as β-hydroxymyristic (14:0 (3-OH)), β-hydroxypalmitic (16:0 (3-OH)), and β-hydroxystearic (18:0 (3-OH)) acids (Figure 3 and Table 1). We also detected ribose (Rib), rhamnose (Rha), galactose (Gal), and glucose (Glc). Repeated RNase digestion and dialysis did not alter the Rib content, indicating that the ribose is an integral part of the LPS and not due to RNA contamination. Other glycosyl residues were rhamnose (Rha), galactose (Gal), glucose (Glc), and *N*-acetylglucosamine (GlcNAc).

Alongside the purified Lcr LPS, standards were run for each of the carbohydrates. From this, we were able to establish response factors for each monosaccharide present in the LPS. This allowed us to quantitate the individual carbohydrate residues (Table 1).

### 2.3. Composition Analysis of SEC Purified Fractions from Lcr

Two fractions of the eluted Lcr released *O*-polysaccharide (see Figure 2, F1 and F2) were collected, and glycosyl composition analysis was performed as above. The analysis revealed that the larger molecular weight peak material (eluting between 6–8 min, F1) was enriched in ribose, rhamnose and N-acetylglucosamine (Table 2). This indicates *O*-polysaccharide polymers containing these residues are enriched in the higher molecular weight portions of the Lcr LPS. Also, the presence of Kdo in the higher molecular weight F1fraction indicates at least some of the LPS core is attached to higher molecular weight *O-*polysaccharides. The lower molecular weight material (eluting between 8 and 10 min, F2) was enriched in Kdo and glucose (Table 2), indicating these residues are enriched in the lower molecular weight core of the Lcr released *O*-polysaccharide. Amounts of galactose were similar between the two fractions, indicating that this residue is present in both the lower molecular weight core and higher molecular weight *O*-polysaccharide repeating units in similar amounts.

Importantly, the composition analysis of the collected fractions did not reveal the presence of any β-hydroxy fatty acids. This indicates that none of the collected oligosaccharide fractions are the result of aggregated LPS still containing lipid A.

### 2.4. Composition Analysis of the Released Lipid A

The composition analysis of the released lipid A (Figure 4) gave the same major fatty acids as those in the intact LPS (Figure 3), namely 14:0 (3-OH), 16:0 (3-OH), and 18:0 (3-OH). In addition, the saturated fatty acyl residue C16:0, the unsaturated residue C18:1, and a very-long-chain fatty acid (VLCFA), 27-hydroxyoctacosanoic acid (28:0 (27-OH)), were observed. The presence of VLCFA is consistent with its identification in the LPS from other members of the *Rhizobiaceae* family [14]. The presence of C18:1 is likely the result of contaminating phospholipids. In addition to the major peaks, several minor fatty acyl peaks were also observed, such as C20:1 and C20:0, etc. (Figure 4).

Amide-linked fatty acids are more resistant to methanolysis than ester-linked acyl chains [23]. This allows us to distinguish amide-linked from ester-linked fatty acids by performing a milder methanolysis on the released lipid A. Two peaks in the resulting chromatogram had mass spectra that were consistent with acyl-GlcN conjugates. Both of these contained 14:0 (3-OH) acyl chains, indicating that the major amide-linked fatty acyl residue was 14:0 (3-OH), and the other observed fatty acyl residues were ester-linked.

### 2.5. Linkage Analysis of Lcr LPS

Linkage analysis of the purified LPS from Lcr was also performed (Figure 5 and Table 3). The results of this analysis were consistent with the composition data showing ribose, N-acetylglucosamine, and galactose to be the main residues detected with smaller amounts of rhamnose and glucose also present. The ribose detected in the sample was shown to be exclusively in the furanose form. The N-acetylglucosamine residues are sourced from both the polysaccharide portion of the LPS as well as the lipid A backbone. There were also several peaks whose mass spectra did not correspond to carbohydrates and are therefore marked with asterisks.

### 2.6. Mass Spectrometric Analysis of the Released Lipid A

MALDI-TOF mass spectra of the intact lipid A were acquired in both positive and negative mode (Figure 6). Table 4 lists the proposed compositions of the lipid A species for each major ion. The positive ion spectra (Figure 6, top) show two ion clusters differing by 28 Da; with a major ion of [M+Na]^+^ *m/z* 1746.33 and a minor ion of [M+Na]^+^ *m/z* 1774.35. The 28 Da difference in this cluster of ions is consistent with the latter ion containing a fatty acyl residue that is two carbons longer than the former ion. Figure 6 also shows molecular species at *m/z* 1848.3 and 1876.33; an increase in 102 Da over the 1746.33 and 1774.35, respectively. These ions are [MNa+Na]^+^ ions of lipid A versions of the monophosphorylated 1746.33 and 1774.35 ions; i.e., 22 Da due to additional sodium and 80 Da due to phosphate. The presence of phosphate in the minor peaks of the sample suggested the use of negative mode (Figure 6, bottom). This spectrum shows ions at *m/z* 1802.97 and 1830.99. These are the [M-H]^−^ ions of the monophosphorylated ions, corresponding to the *m/z* 1848.30 and 1876.33 lipid A species observed in the positive ionization mode. We also see in the negative ionization MS a minor ion at m/z 1926.96. This lipid A species could be due to the addition of a phosphoethanolamine group to the *m/z* 1802.97 species.

The analysis of the lipid A from Lcr reveals the major ion species to consist of a pentaacylated GlcN disaccharide backbone. Two β-hydroxylmyristic acid residues are amide linked with one containing a linked 27-hydroxyoctacosanoic acid as an acyloxyacyl residue. Two 16:0 (3-OH) residues are ester-linked. Our analysis did not show any evidence of a proximal glucosamine oxidation which is found in the LPS of some members of the *Rhizobiaceae* family [14]. However, the monophosphorylated species leaves open the possibility that the native lipid A contains a phosphate on the reducing glucosamine. Phosphate at this position is typically hydrolyzed during lipid A purification using mild acidic conditions. So, although the phosphorylated lipid A species is the minor ion in the MALDI-TOF analysis, it is possible it is the major species of lipid A present in the native sample.

### 2.7. NMR Analysis of the O-polysaccharide from Lcr

The 1-D proton NMR spectrum of the polysaccharide released from Lcr LPS after removal of lipid A by mild acid hydrolysis is shown in Figure 7. The anomeric region of the spectrum featured 7 major and about 10–12 minor resonances. The proton NMR spectrum was also characterized by the presence of a complex region between 4.4 and 3.5 ppm due to the remaining glycosyl ring protons, N-acetyl methyl signals at 2.04 ppm, a few smaller signals between 2.2 and 1.8 ppm, corresponding to the H-3 protons of Kdo, and a strong multiplet signal around 1.28 ppm, belonging to H-6 of several Rha residues.

The residues belonging to the major anomeric signals were labeled with letters and assigned using a set of 2-dimensional ^1^H-^1^H- and ^1^H-^13^C-NMR correlation experiments, including ^1^H-^1^H-COSY, TOCSY, and NOESY and ^1^H-^13^C-HSQC and HMBC.

Residues A, B, C, E, and G contained anomeric protons resonances of 5.50, 5.42, 5.40, 5.33, and 5.22 ppm (Table 5), respectively. According to the HSQC spectrum, these anomeric protons were attached to carbons resonating between 109.2 and 111.0 ppm (Table 5), indicating that all of these residues were in the furanose ring form [24]. This was further confirmed by HMBC correlations between H-1 and C-4 for each residue (Figure 8 and Figure 9) and linkage analysis (Figure 5) which found Rib to be in the furanose form. The ^3^J_H1-H2_ coupling constant for each ribofuranosyl (Ribf) residue was too small to measure, indicating β-anomeric configurations [25]. Residues A, B, and C had carbon-2 resonances of 84.0, 83.7, and 83.7 ppm respectively (Table 5), which is 5.9 and 5.6 ppm downfield from C-2 of unsubstituted β-Ribf [26], identifying these as 2-linked-β-Ribf-1→ residues. Conversely, C-2 of residues E and G appeared slightly upfield of referenced unsubstituted β-Ribf, and their C-3 resonances appeared at 81.9 and 82.3 ppm (Table 5), 8.6 and 9.0 ppm downfield of unsubstituted β-Ribf, indicated that residues E and G were 3- β-Ribf-1.

The chemical shifts of residues P (H-1 at 4.497 ppm) and Q (H-1 at 4.493 ppm) were very similar. The anomeric signals of these two residues could only be distinguished in the 2D NOESY and HMBC spectra (by their correlations to different neighboring residues, see below) but were not resolved in the proton spectrum. Both residues were characterized by a TOCSY pattern with 3 cross-peaks correlated with H-1. The COSY spectrum revealed the sequence of these peaks within the monosaccharide ring and assigned H-2 and H-3 at 3.57 and 3.77 ppm for both residues P and Q, and H-4 at 4.03 ppm for residue P or 4.05 ppm for residue Q, respectively. This peak pattern is consistent with these residues having a β-galactopyranose [24] configuration. Compared with monomeric β-Gal*p* [27], the chemical shift of C-4 of both P and Q displayed a strong α-effect of about +7 ppm, whereas C-3 exhibited a weak β-effect of −0.2 ppm. This suggested the two residues were 4-β-Galp-1→.

The sequence of the monosaccharide residues A, B, C, E, G, P, and Q was determined by inspecting the ^1^H-^1^H-NOESY and ^1^H-^13^C-HMBC NMR spectra for inter-residue correlations. Here, two different polymers were observed. The simpler of the two consisted of a disaccharide repeating unit of residues E and Q. The HMBC showed clear correlations between the anomeric proton of Q at 4.49 ppm and the C-3 of E at 81.9 ppm. Similarly, the anomeric proton of E at 5.33 ppm was correlated to the C-4 of Q at 78.5 ppm. The NOESY spectrum showed equivalent results, with correlations between the anomeric proton of residue E at 5.33 and H-4 of residue Q at 4.05 ppm. A similar correlation was present between the anomeric proton of Q at 4.49 ppm and the H-3 of residue E at 4.21 ppm (Figure 9). No other inter-residue correlations were found connecting these two residues with any of the other major residues in the sample, although the terminal β-Gal residue W was found at the non-reducing end of this chain. Taken together, these results led to the determination that these two residues were part of a disaccharide repeat with the sequence 3-β-Ribf-1→4-β-Galp-1→ (E-Q).

Connectivities between the remaining major residues were further examined. The HMBC spectrum showed inter-residue connections between H-1 of B and C-2 of A (5.42/84.0 ppm), between H-1 of A and C-4 of P (5.50/78.5 ppm), between H-1 of P and C-3 of G (4.50/82.3 ppm), between H-1 of G and C-2 of C (5.22/83.7 ppm), between H-1 of C and C-2 of B (5.41/83.7 ppm). The NOESY spectrum confirmed these connectivities with cross-peaks correlating H-1 of B with H-2 of A (5.42/4.22 ppm), H-1 of A with H-4 of P (5.50/4.03 ppm), H-1 of P with H-3 of G (4.50/4.31 ppm), H-1 of G with H-2 of C (5.22/4.21 ppm), and H-1 of C with H-2 of B (5.41/4.20 ppm). These correlations indicated the pentasaccharide repeat sequence B-A-P-G-C or [2-β-Ribf-1→2-β-Ribf-1→4-β-Galp-1→3-β-Ribf-1→2-β-Ribf-1→]_n_. Two minor terminal O-acetylated ribose residues, S and T make up the non-reducing end of this polymer, S being acetylated on O-3 and T on O-2.

Among the minor anomeric signals were two more that corresponded to the residues of a disaccharide repeating unit. Residue M with an anomeric proton chemical shift of 4.91 ppm was identified as α-rhamnopyranose by its downfield H-2 (4.08 ppm) and H-5 (4.05 ppm) and upfield H-6 (1.26 ppm). An α-effect of +5.3 ppm and a β-effect of −3.6 ppm [28] indicated glycosylation on O-3. Residue L was assigned as α-GlcNAc because of its ^3^J_H1,H2_ coupling constant of 3.8 Hz and its upfield H-4 (3.68 ppm) and C-2 (56.6 ppm) resonances. An α-effect of +6.9 ppm of C-4 (80.1 ppm) relative to H-4 of unsubstituted GlcNAc [28] indicated glycosylation in this position. The connections between residues M and L were established by HMBC cross-peaks at 4.91/80.1 ppm and 5.01/78.2 ppm and by NOESY cross-peaks at 4.91/3.68 ppm and 5.01/3.76 ppm, showing a disaccharide repeat of 3-α-Rhap-1→4-α-GlcpNAc-1→ (M-L). This repeating unit structure is identical to that of the *O*-polysaccharide of the LPS from *Actinobacillus actinomycetemcomitans* (OMZ 534) serotype e [29].

The residue labeled R contained a pair of resonances at 1.86 and 2.07 ppm arising from a methylene group, as evidenced by its negative intensity in the multiplicity-edited HSQC spectrum. These peaks are characteristic of the Kdo H-3 protons consistent with the observation of Kdo during glycosyl composition analysis (Figure 3). This residue links the core oligosaccharide to lipid A in the LPS. Downfield C-1 (179.2 ppm) and upfield C-2 (99.2) chemical shifts indicated α-anomeric configuration of this residue [30,31], and a downfield displacement of the C-5 chemical shift (+9.4 ppm) [32] gave evidence of it being glycosylated on O-5.

The anomeric peaks of residues O and N contained proton anomeric resonances at 4.51 and 4.65 respectively. The TOCSY spectrum showed three cross-peaks each to these anomeric protons. Following the correlations in the COSY spectrum gave H-2, H-3, and H-4 chemical shifts of 3.67, 3.72, and 4.00 ppm for residue O and 3.60, 3.78, and 4.02 ppm for residue N. These peak patterns are consistent with these residues having a β-galactopyranose [24] configuration. Compared with unsubstituted β-Gal*p* [27], the chemical shift of C-4 of residue N and C-3 of residue O displayed strong α-effects of about 9.0 and 8.8 ppm respectively, suggesting residue N was 4-β-Galp-1→ and residue O was 3-β-Galp-1→.

The anomeric signal of residue F resonated at 5.32/111.1 in the HSQC spectrum, indicating furanose form. Since ribose was the only furanose found in the linkage, residue F could be identified as Ribf. The lack of a measurable ^3^J_H1-H2_ coupling constant indicated it was in the β-anomeric configuration [25]. The chemical shift of C-3 for this residue was 82.0 ppm, a downfield displacement of over 8 ppm from unsubstituted β-Ribf, identifying residue F as 3-β-Ribf-1→.

The TOCSY spectrum allowed assignment of residues K and J as rhamnosyl residues based on the cross-peaks of H-5, H-4, H-3, and H-2 with H-6 at 1.28 and 1.30 ppm, respectively. These latter, upfield proton chemical shifts are characteristic of the C-6 methyl of 6-deoxyhexose residues, and rhamnose is the only 6-deoxyhexose residue present in these samples, as observed in the composition and linkage analyses. H-5 of both of these residues resonated above 3.8 ppm and C-5 around 72 ppm, values are consistent with K and J having an α-anomeric configuration [28]. The chemical shifts of C-3 of residues K and J at 80.9 and 82.7 ppm respectively, showed large α-effects of 9.9 and 11.7 ppm from unsubstituted α-Rhap, identifying residues K and J as 3-α-Rhap-1→.

The H-1 of residue I resonated at 5.15 ppm. The TOCSY spectrum showed four cross-peaks correlated to the anomeric proton of this residue. Sequencing these peaks in the COSY spectrum identified H-2, H-3, H-4, and H-5 as having chemical shifts of 3.59, 3.94, 3.71, and 4.24 ppm, a pattern consistent with α-Glcp. A ^3^J_H1-H2_ coupling constant of 4 Hz confirmed α-anomeric configuration. The chemical shift of C-4 for this residue was 80.6 ppm, which is a downfield displacement of over 9 ppm from unsubstituted α-glucopyranose, identifying residue I as 4-α-Glcp-1→.

The HMBC spectrum showed correlations between H-1 of residue I and C-5 of R (5.15/78.1 ppm), between H-1 of O and C-4 of I (4.51/80.6), between H-1 of K and C-3 of O (5.02/82.8 ppm), and between H-1of J and C-3 of K (5.05/80.9 ppm). The NOESY spectrum showed these same four residues to have corresponding correlations between their anomeric protons and ring protons H-5 on residue R at 4.09 ppm, H-4 of residue I at 3.71 ppm, H-3 of residue O at 3.72 ppm, and H-3 of residue K at 3.91 ppm respectively. These correlations indicated the residue sequence J-K-O-I-R or 3-α-Rhap-1→3-α-Rhap-1→3-β-Galp-1→4-α-Glcp-1→5-α-Kdop.

The HMBC and NOESY spectra furthermore showed correlations between H-1 of residue N and C-3 and H-3 of J (4.65/82.7 ppm and 4.65/4.00 ppm). A correlation between H-1 of F and H-4 of N (5.32/4.02 ppm) was also found, allowing the conclusion that the disaccharide 3-β-Ribf-1→4-β-Galp-1→ is attached to the non-reducing end of the core pentasaccharide, forming the sequence F-N-J-K-O-I-R. It is likely that the E-Q- repeat is attached to the non-reducing end of F, although the similarity of the chemical shifts of F with those of E and the associated peak overlap and, thus, precluded confirming this linkage. The complete core oligosaccharide consists of the pentasaccharide polymer 3-α-Rhap-1→3-α-Rhap-1→3-β-Galp-1→4-α-Glcp-1→5-α-Kdop-1 with residues F and N comprising the first polysaccharide disaccharide repeat linking the *O*-polysaccharide to the core.

Two additional non-reducing end residues, U and V, constitute the termini of core structures, truncated at the sixth (i.e., U replacing N) or fifth (i.e., V replacing J) positions, respectively, from Kdo. These, together with the three other terminal residues, S, T, and W, helped define the different LPS/LOS species isolated from Lcr. No reducing end residues other than Kdo were detected, indicating that the five non-reducing end sugars each represented an O-chain-core lipopolysaccharide (LPS) or core only lipooligosaccharide (LOS) and that each corresponded to a species that included one or more of the different sequences described above. Thus, structures 1 and 2 contained the F-N-J-K-O-I-R core, a number of E-Q repeats, and several B-A-P-G-C units. Structure 3 comprised the F-N-J-K-O-I-R core and a number of E-Q repeats, and structures 4 and 5 consisted of different lengths of core only (Figure 10). The number of O-chain repeats B-A-P-G-C and E-Q was estimated from the average intensity of their anomeric proton signals.

The intensity of the terminal residues S, T, U, V, and W was used to estimate the molar amounts of the five structures present. Then, the expected intensities of all residues were back-calculated using the numbers of residues in, together with the relative molar amounts of each of the five structures (Table 6). A comparison of the calculated with the observed intensities gave reasonably good agreement, confirming the accuracy of the proposed structures. Knowledge of the monosaccharide composition thus obtained of each structure also gave their molecular weights and, together with the molecular weight determined above of the lipid A (~1.8 kDa), the molecular weights of the corresponding LPS/LOS species (Figure 10). The five structures fall into three size categories: LPS structures 1 and 2 are fairly large (~11 kDa), LOS structures 4 and 5 are small (~3 kDa), and structure 3 is of intermediate size (~5 kDa). The abundance ratio of the three categories is about 2:2:1. These observations agree with the DOC-PAGE analysis (Figure 1), which showed two intense bands near the bottom and a weaker band between them. These findings also agree with the SEC results from the delipidated polysaccharides (Figure 2). The gel also showed a very faint ladder of much higher molecular weight. This most likely corresponds to the 3-α-Rhap-1→4-α-GlcpNAc-1→ (M-L) polysaccharide, of which no terminal residues were found as would be expected for a very large polysaccharide.

## 3. Discussion

*Liberibacter crescens* remains the only axenically cultured representative of the *Liberibacter* genus. Lcr serves as a genetically tractable model organism for the uncultured plant pathogens associated with several economically important diseases such as citrus greening, potato zebra chip, vein greening and psyllid yellows in tomato, yellows decline in carrots, and vegetative disorders in fennel, celery, parsnip, and parsley [19]. This work describes the structural characterization of Lcr LPS. Composition, mass spectrometric, and NMR analyses showed that the major lipid A species of Lcr LPS consists of a pentaacylated 6-linked GlcN disaccharide backbone with two amide-linked 14:0 (3-OH) residues, two ester-linked 16:0 (3-OH) residues, and one 27-hydroxyoctacosanoic acid linked as an acyloxylacyl residue (Figure 11). Notably, the major lipid A backbone species (as evidenced in the MALDI) lacks the typical di-phosphate moieties, while also not replacing the negative charge by the addition of galacturonic acid or oxidation of the proximal glucosamine as is the case for a number of other members of the *Rhizobiaciae*. However, the mass spectrometric analysis showed a minor lipid A species that does contain a single phosphate group. Given the phosphate present on the proximal glucosamine is particularly labile, it is possible the native Lcr LPS exists as a 1-monophosphorylated species, and the results we present here of a lipid A lacking phosphate are an artifact of the hydrolysis conditions employed to isolate the lipid A.

An extensive 2D NMR analysis of the lipid A-released polysaccharide allowed us to characterize the core oligosaccharide with an attached ribose and galactose containing disaccharide repeat, as well as two additional short polysaccharides, one of them a ribose and galactose containing pentasaccharide repeat and the other a rhamnose and *N*-acetylglucosamine containing disaccharide repeat (Figure 10). These two latter polysaccharides may either be additional O-antigens belonging to different LPS species, or they may be attached to the Rib-Gal disaccharide repeat, which then could be designated as outer core. The NMR data were not able to confirm a covalent connection between the three polymers. However, the peak intensities of the residues making up these polymers were only 2–4 times greater than those of the residues making up the core oligosaccharide. If the polymers are not connected to one another, it is hard to explain the results from the SEC and DOC-PAGE analysis indicating some of the Lcr polysaccharides are in excess of 3.9 kDa.

Structural elucidation of the Lcr LPS has given us valuable insight into the biology of the *Ca*. Liberibacter spp. All structural genes of the Raetz pathway [33] for lipid A biosynthesis (except for *lpxM* encoding myristoyltransferase, EC:2.3.1.243) have been annotated in the genomes of Lcr and pathogenic *Ca*. Liberibacter spp. Pentaacylated Lcr lipid A (28:0 (27-OH)) displays significant similarity to lipid A of phylogenetically related *Sinorhizobium meliloti* (28:0 (27-OH)) and *Brucella abortus* [28:0 (27-OH(βOmeC4:0))] [34]. *S. meliloti* genes *lpxXL* (SMc04268) and *acpXL* (SMc04278) encode a lipid A C28 acyltransferase (EC:2.3.1.241) and an acyl carrier protein required for the biosynthesis of VLCFA-modified lipid A. Homologs for both these genes are present in Lcr (B488_RS04675 and B488_RS04700, respectively) but absent in all pathogenic (and uncultured) ‘*Ca*. Liberibacter’ spp. Multiple attempts to inactivate Lcr *lpxXL* by marker interruption have remained unsuccessful to date [19,35], leading to the speculation that the uncultured ‘*Ca*. Liberibacter’ spp. may require VLCFA-modified lipid A for free-living growth in culture. The lack of VLCFA-modified lipid A has been associated with impaired symbiosis and compromised outer membrane permeability increasing sensitivity towards pH, detergents, peptide antibiotics, osmotic pressure, and desiccation [21]. Genetic analyses predict the *Ca*. Liberibacter spp. LPS to be tetraacylated as opposed to the pentaacylated structure seen in Lcr. Notably, in *E. coli*, tetraacylated lipid A precursors are not as efficiently translocated to the outer membrane as are penta- or hexa-acylated lipid A species leading to temperature-dependent growth defects [36,37].

Structurally, deoxy-hexoses and methylated sugar residues tend to be abundant in the *O*-polysaccharides from members of the *Rhizobiaceae* family [14] resulting in relatively hydrophobic *O*-polysaccharides. This is in contrast to the more polar residues contained in the core oligosaccharide of *Rhizobiaceae* family LPS. The core and *O*-polysaccharide structure presented here reverses this. The structural analyses presented here show two of the three *O*-polysaccharide structures are comprised of repeating ribofuranose and galactose moieties instead of deoxyhexose residues. Further, the composition analysis does not suggest the presence of methylations on any of the monosaccharides. On the other hand, deoxy-hexoses residues (rhamnose) are proposed to constitute the core oligosaccharide, along with a single acidic Kdo residue. The significance of this inverted *O-polysaccharide* structure is unknown, and given the tendency for *O*-polysaccharides to be highly variable between strains, this structure is unlikely to be shared between other members of the *Ca*. Liberibacter spp. However, the core oligosaccharides do tend to be more conserved. Thus, it is likely other *Ca*. Liberibacter spp. may share the core oligosaccharide structural features presented here.

In host–pathogen interactions, *O*-polysaccharides serve as an important barrier, providing protection from host-associated defense mechanisms, as well as environmental toxins. At the same time, *O*-polysaccharides play important roles as virulence factors, with the large *O*-polysaccharides generally being present in virulent bacteria. Avirulent bacteria have either reduced size or completely lack the *O*-polysaccharide. For example, in *Salmonella enteritidis*, it was shown that virulent bacteria had a higher number of *O*-polysaccharide repeating units, and thus a larger molecular size, compared to avirulent strains [37]. Similarly, it has been shown that the rough LPS mutants of the intracellular *Brucella* spp. lack *O*-polysaccharide and tend to be avirulent [38]. *O*-polysaccharide modifications have also been shown to be important in modulating chronic vs. acute infections in *P. aeruginosa*, it was reported that the shift from acute to chronic infection resulted in clinical isolates containing LPS lacking *O*-polysaccharide [39]. Thus, while the reduced size of the *O*-polysaccharide reported here for Lcr would likely result in the bacteria being more sensitive to external plant defensive molecules, its restricted lifestyle likely minimizes these risks. At the same time, the reduced *O*-polysaccharide size could serve as an important mitigating factor in the bacteria’s interactions with its host and its ability to establish long-lasting infections.

## 4. Materials and Methods

### 4.1. Growth of Lcr Cells

*Liberibacter crescens* BT-1 cells (ATCC^®^ BAA-2481) cultures were supported on BM7A medium. The media consisted of 3.75 g KOH in 550 mL water, 20 g of *N*-(2-acetamido)-2-aminoethanesulfonic acid (ACES) buffer, and 2 g α-ketoglutaric acid, pH = 6.9. This was followed by the addition of 150 mL filter-sterilized fetal bovine serum (HyClone Laboratories, Logan, UT, USA) and 300 mL modified Grace’s insect culture medium (TNM-FH; HyClone Laboratories) [33]. Due to the extremely slow growth rate and fastidious nature of Lcr, it was critical to initiate a starter culture in 2 mL and expand the culture volume stepwise to 10, 25, 100 mL allowing 3-days growth between each stage. A small culture aliquot was PCR-verified for purity at each successive transfer as previously described [19]. Finally, 10 L of the bacterial culture was grown for five days with gentle shaking at 150 rpm at 28 °C and harvested at midlog phase (Abs_600_ = 0.6). The bacteria were centrifuged for 15 min at 5000× *g*, 4 °C. The bacterial pellet was resuspended and washed in pre-chilled, sterile water and centrifuged again.

### 4.2. Isolation of LPS

LPS was extracted using hot phenol and water as previously described [40]. A solution of the dried cell pellet in deionized (DI) water was made, 10 mL of DI water per gram of pellet, and an equivalent amount of 90% phenol was added. The solutions were preheated to 68 °C, mixed and stirred for 20 min at 68 °C, and then placed in an ice bath to cool. The chilled solution was then centrifuged at 5000× *g* for 20 min, and the aqueous layer was collected. An amount of DI water (heated to 68 °C) equivalent to what was removed was added to the phenol layer (heated to 68 °C), and the process was repeated twice more. The three water layers were combined and dialyzed using a 10 kDa dialysis membrane. The water-extracted, dialyzed, and lyophilized LPS was digested with RNAse and DNAse (Sigma) in 50 mM MgCl buffer (pH = 7.5) overnight at 37 °C (10 mg of LPS/mL solution, 3.5 U RNAse/mL, 3.5 U DNAse/mL). This was followed by proteinase K digestion (Sigma) (20 U Proteinase/mL). The sample was then dialyzed as before and further purified by ultracentrifugation (100,000× *g*, 16 h). The ultracentrifugation pellet represented the purified LPS and was used for the analysis described below.

### 4.3. Deoxycholate-Polyacrylamide Gel Electrophoresis (DOC-PAGE)

LPS was analyzed by PAGE using an 18% acrylamide gel with deoxycholic acid (DOC) as the detergent [41]. The LPS bands were visualized by fixing the gel in the presence of a silver stain [42].

### 4.4. Composition Analysis of the LPS and Lipid A by TMS Methyl Glycosides

For the glycosyl composition analysis, per-*O*-trimethylsilyl (TMS) derivatives were generated from the methyl glycosides produced after hydrolysis of the sample as described previously [43].

The GC ramp temperature program was as follows. Starting with an initial temperature of 80 °C, the oven was ramped at a rate of 20 °C per minute until the temperature reached 140 °C, which was held for 2 min. After 2 min, the GC was ramped at a rate of 2 °C per minute until it reached 200 °C. Finally, it was ramped at a rate of 30 °C per minute until the GC reached a final temperature of 250 °C, where it was held for 5 min.

For the mild lipid A composition analysis, the procedure was identical except the methanolysis time was decreased to 4 h, and the re-*N*-acetylation step was skipped to prevent overlap of N-acetylglucosamine peaks with minor fatty acid peaks.

### 4.5. Isolation of Lipid A

Lipid A was released from the LPS (0.5 mg) by mild hydrolysis in aqueous 1% acetic acid (250 μL) at 100 °C for 1 h. Lipid A precipitate was then collected via centrifugation (2000× *g*, 15 min.). The precipitated lipid A was washed with water twice more and centrifuged to remove any residual polysaccharide. The precipitated lipid A was used for mass spectrum analysis and the released polysaccharide component was used for LC and NMR analysis.

### 4.6. MALDI-TOF Mass Spectrometric Analysis of the Lipid A

The released lipid A was dissolved in 50 μL methanol, and an aliquot of 2 μL of the resulting solution was mixed with an equal volume of matrix solution (20 mg/mL 2,5-dihydroxybenzoic acid (DHB) matrix solution in water:methanol 1:1 was used for the positive ion spectra and 20 mg/mL 2′,4′,6′-trihydroxyacetophenone (THAP) matrix solution in water:acetonitrile 1:1 was used for negative ion mode). The sample was spotted on the MALDI plate and analysis was performed on the sample using an Applied Biosystems 5800 MALDI in the positive or negative ionization mode.

### 4.7. NMR Analysis of the Released O-polysaccharide

The *O*-polysaccharide preparation (~1.5 mg) was dissolved in 400 μL D_2_O (99.96% D, Cambridge Isotope Laboratories), 1 μL 50 mM DSS-d6 (Cambridge Isotope Laboratories) in D_2_O was added and the mixture was lyophilized twice. Following second lyophilization, the sample was dissolved in 40 μL D_2_O (99.96% D) and transferred into a 1.7 mm NMR tube. NMR data were acquired at 25 °C on a Bruker NEO (^1^H, 799.71 MHz) spectrometer equipped with a 1.7 mm cryoprobe. The 2D COSY, TOCSY, and NOESY spectra were acquired with 8 transients per 320 increments and ^1^H spectral widths of 5556 Hz. The TOCSY spectrum was acquired with a mixing time of 80 ms and the NOESY spectrum with that of 70 ms. The HSQC spectrum was acquired with 8 transients per 400 increments, and the HMBC with 32 transients per 300 increments. ^13^C spectral width was set to 14,077 Hz for both heteronuclear experiments. ^1^H and ^13^C chemical shifts were referenced to the respective DSS signals at 0.0 ppm. All spectra were processed and analyzed with MestreNova or CCPN Analysis [44].

### 4.8. SEC Analysis of the Released O-polysaccharide from Lcr LPS and E. coli EH100 Mutant

The *O*-polysaccharide was released from the Lcr LPS and the *E. coli* EH100 Ra mutant as described above. The dried *O*-polysaccharide was dissolved in 20 mM ammonium acetate buffer pH = 5.5, (1 mg/mL), and 100 μL was injected into an Agilent 1260 HPLC. Separation was achieved using a Superdex Peptide 10/300 gL column (GE). The sample was eluted with 20 mM ammonium acetate at 1 mL/min in isocratic condition and detected with the refractive index detector (Agilent).

### 4.9. Glycosyl Linkage Analysis of the Released O-polysaccharide

Glycosyl linkage analysis consisted of permethylation, depolymerization, reduction, and acetylation steps. This procedure generated partially methylated alditol acetates (PMAAs), which could be analyzed by gas chromatography-mass spectrometry (GC-MS). The procedure was performed as previously described [45].

## Figures and Tables

**Figure 1 ijms-22-11240-f001:**
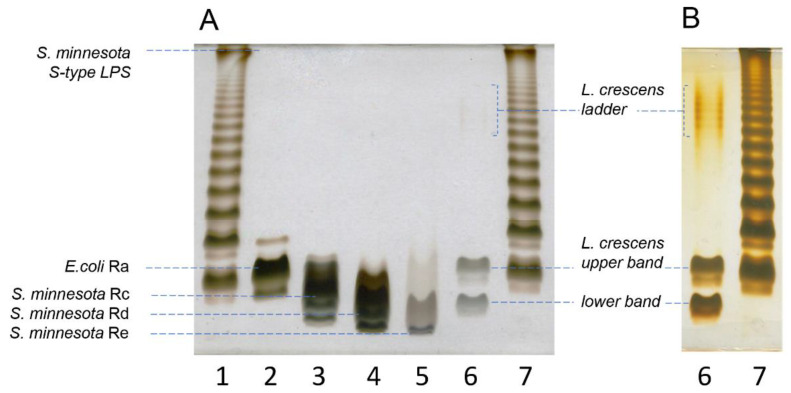
Comparative DOC-PAGE analysis of the LPS from Lcr. Panel (**A**): LPS from Lcr (Lane 6, 0.5 µg) was run alongside LPS of *Salmonella enterica* serovar Minnesota S-strain LPS (Lanes 1 and 7, 2 µg); *E. coli* EH100 (Ra mutant) (Lane 2, 0.5 µg), *S. minnesota* R5 (Rc mutant) (Lane 3, 0.5 µg), S. minnesota R7 (Rd mutant) (Lane 4, 0.5 µg) and *S. minnesota* R595 (Re mutant) (Lane 5, 0.5 µg). Panel (**B**): Overexposure of the gel to silver staining allowed for viewing of the faint higher molecular weight bands from Lcr LPS.

**Figure 2 ijms-22-11240-f002:**
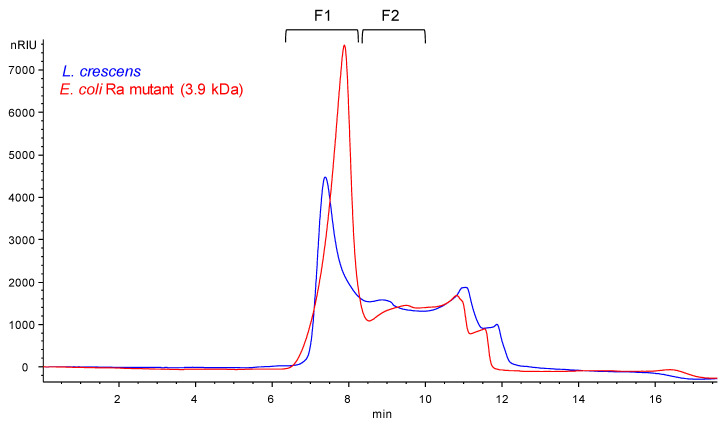
SEC chromatogram of the *O*-polysaccharide released from the Lcr. The chromatogram from Lcr *O*-polysaccharide was overlaid with the core region released from *E. coli* EH100 (Ra mutant, 3.9 kDa). Fractions 1 and 2 (F1, F2) indicate two collected fractions that were used for composition analysis.

**Figure 3 ijms-22-11240-f003:**
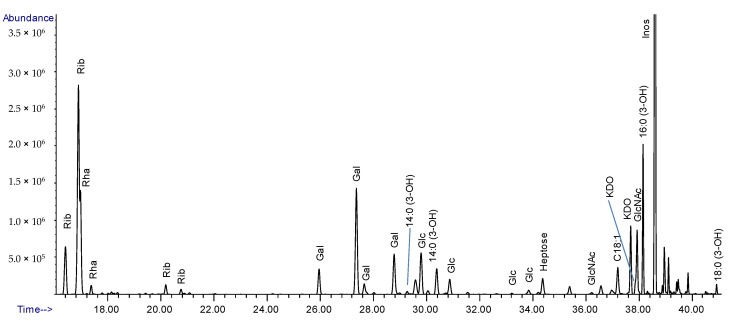
Composition analysis of the LPS from Lcr. GC/MS chromatogram of the fatty acids and carbohydrates released from the purified LPS via acid methanolysis and TMS derivatization.

**Figure 4 ijms-22-11240-f004:**
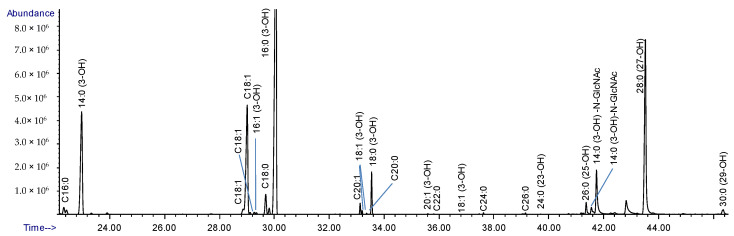
Composition analysis of the isolated lipid A from Lcr. LPS from Lcr was released by mild acid hydrolysis and preparation of TMS methyl glycosides. The fatty acid composition shows residues consistent with lipid A from *Rhizobiaceae* (28:0 (27-OH)), and the presence of 14:0 (3-OH) fatty acid amide linked to glucosamine.

**Figure 5 ijms-22-11240-f005:**
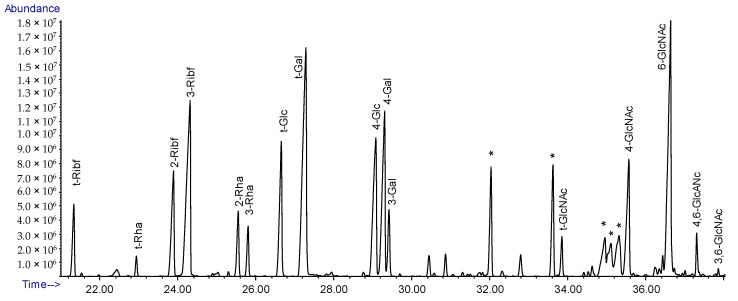
Results of the linkage analysis of the Lcr LPS. Linkage analysis was obtained on the LPS by preparation of partially methylated alditol acetates (PMAA). Asterisks mark unknown non-carbohydrate peaks. The chromatogram also shows the presence of contaminating lipid A (i.e., the presence of 6-linked GlcN) in the sample.

**Figure 6 ijms-22-11240-f006:**
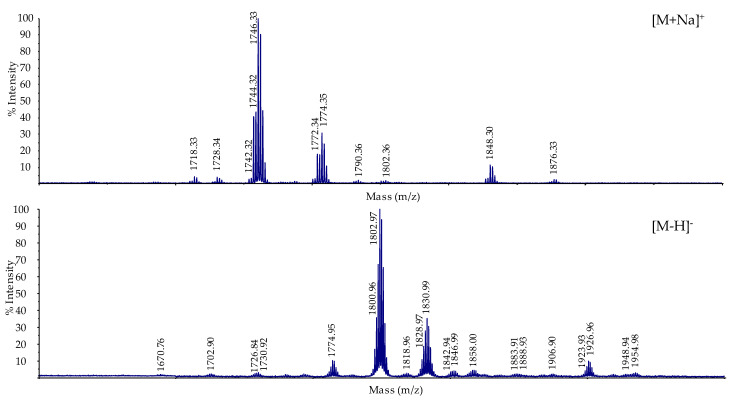
Mass spectrum (MALDI-TOF) of the lipid A released from purified Lcr. LPS. Positive mode is shown on top, negative mode is shown on the bottom. The major ion in the positive mode is a pentaacylated lipid A lacking phosphate with a minor ion representing this species containing a single phosphate. The singly linked phosphate is present as the major ion in the negative mode spectrum. It cannot be ruled out that the major lipid A species in the native LPS is a 1-monophospahte structure, and this phosphate is removed during the mild acid hydrolysis of the lipid A.

**Figure 7 ijms-22-11240-f007:**
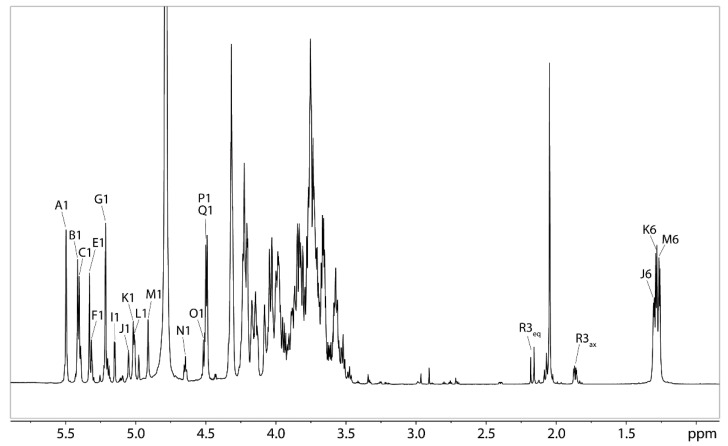
The 1D ^1^H-NMR spectrum obtained from the LPS of Lcr after lipid A removal. The anomeric signals and other distinct signals are labeled. For labels of the other protons/carbons, see Figure 8.

**Figure 8 ijms-22-11240-f008:**
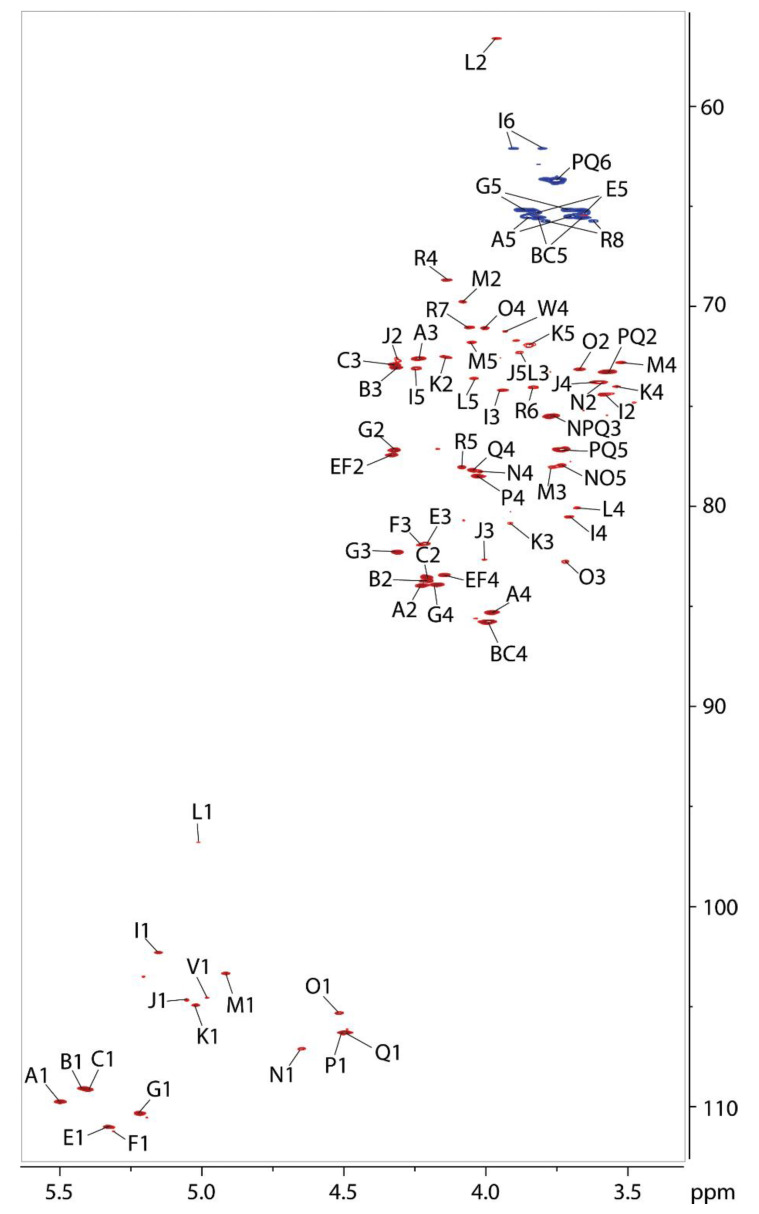
The partial 2-D ^1^H-^13^C-HSQC NMR spectrum of the Lcr polysaccharide. The spectrum was obtained on the polysaccharide released after mild acid hydrolysis of the LPS.

**Figure 9 ijms-22-11240-f009:**
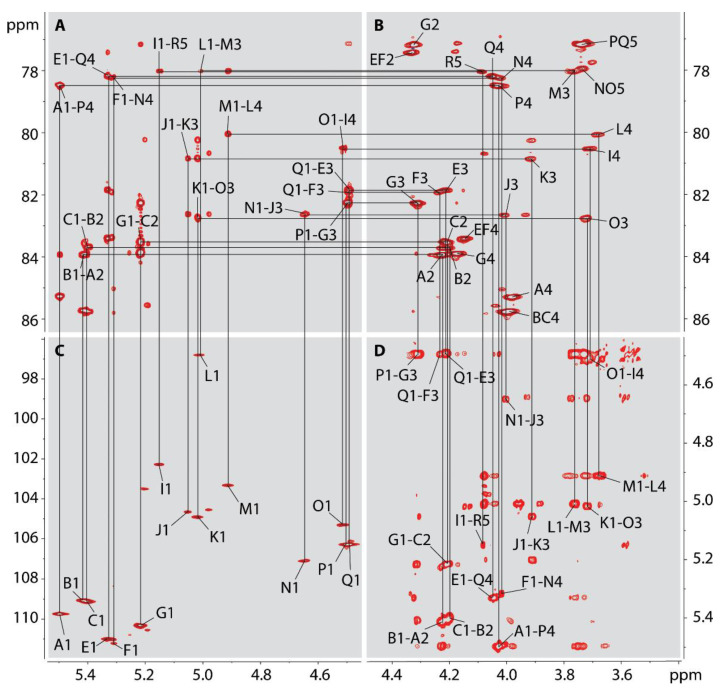
Partial 2-D ^1^H-^13^C-HMBC (**A**), HSQC (**B**,**C**), and ^1^H-^1^H-NOESY (**D**) NMR spectra of the Lcr polysaccharide. The figure shows inter-residue correlations used to determine the monosaccharide sequences of the core and O-antigens.

**Figure 10 ijms-22-11240-f010:**
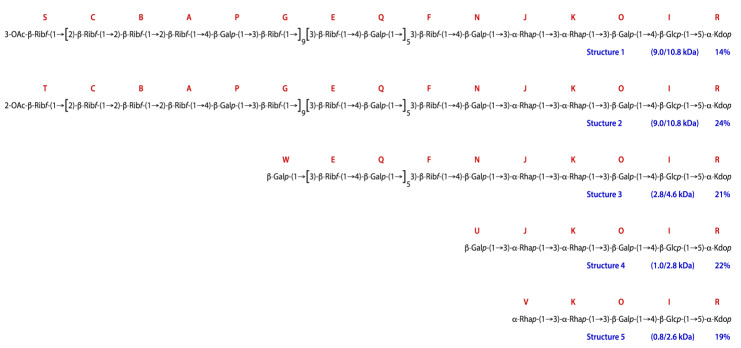
*O*-polysaccharide and core structures elucidated by the NMR analysis. Proposed core-*O*-polysaccharide and core-only carbohydrate structures of delipidated Lcres LPS/LOS. The letters correspond to those listed in Table 5. The molecular weights listed in parentheses refer to those of the depicted polysaccharides and to the corresponding LOS/LPS.

**Figure 11 ijms-22-11240-f011:**
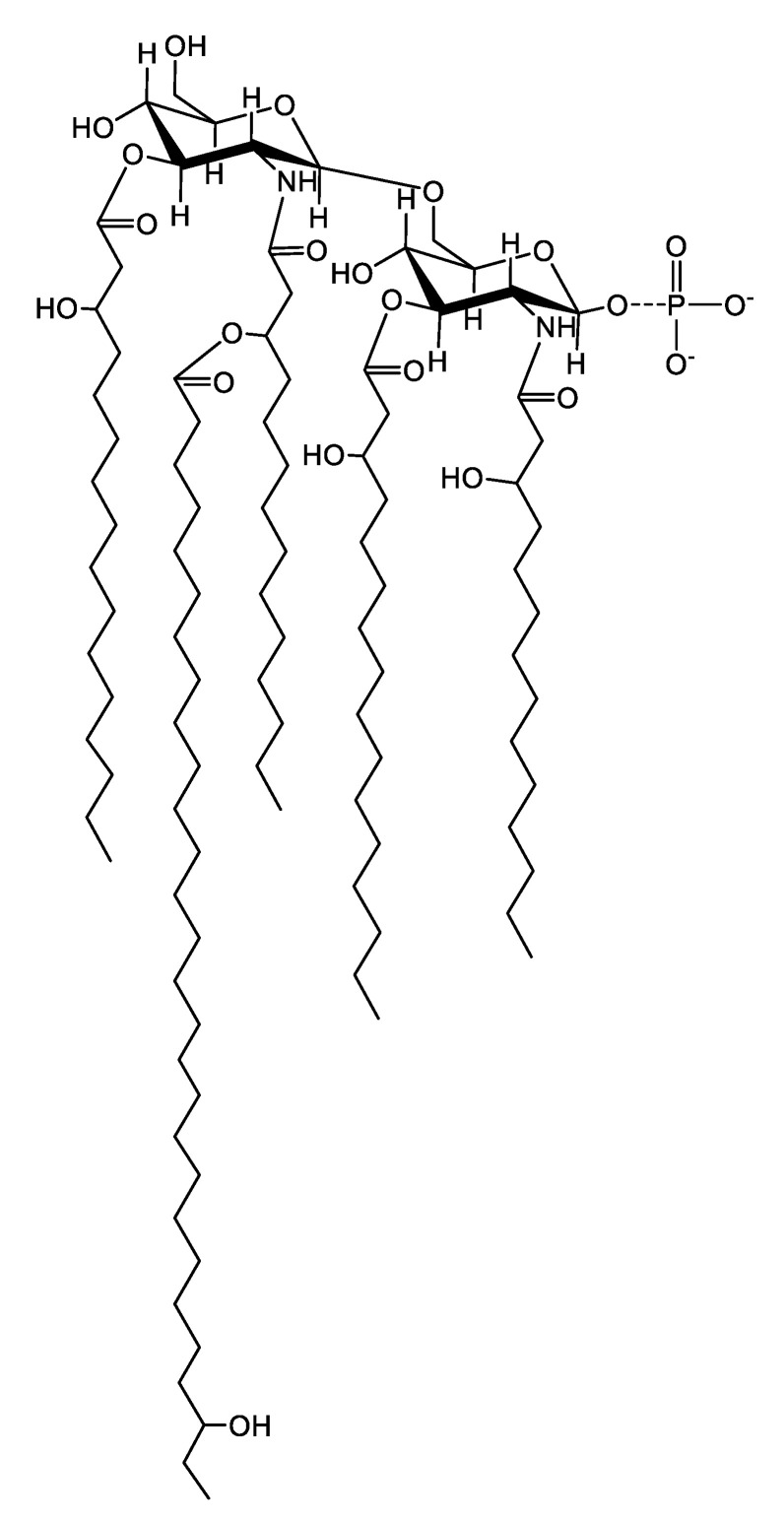
Proposed structure for the major species of Lipid A released from Lcr LPS. The major ion in the positive mode MALDI spectra is a pentaacylated GlcN disaccharide backbone with two amide-linked 14:0 (3-OH) residues, two ester-linked 16:0 (3-OH) residues, and one 27-hydroxyoctacosanoic acid linked as an acyloxyacyl residue. It is possible the lack of phosphate in the major ion is an artifact of the mild acid hydrolysis used for lipid A release and that the native structure is a 1-monophosphoryl species.

**Table 1 ijms-22-11240-t001:** Composition analysis of 200 μg of the intact Lcr LPS.

Glycosyl Residue	Mass (μg)	Mol %
Ribose (Rib)	26.4	43.8
Rhamnose (Rha)	7.8	11.9
Galactose (Gal)	16.6	22.9
Glucose (Glc)	6	8.3
*N*-Acetylglucosamine (GlcNAc)	8.1	13.2
Σ=	65	100

**Table 2 ijms-22-11240-t002:** Ratio (F1/F2) of the detected monosaccharide peak areas from the F1 and F2 fractions after normalizing to inositol. Ratios significantly > 1 indicate the monosaccharides are enriched in the higher molecular weight F1 fraction. Ratios significantly < 1 indicate the residues are enriched in the lower molecular weight F2 fraction.

Glycosyl Residue	Ratio F1/F2
Ribose	3.13
Rhamnose	2.37
Galactose	1.14
Glucose	0.67
3-Deoxy-*D*-manno-oct-2-ulosonic acid	0.17
*N*-Acetylglucosamine	1.97
Inositol	1.00

**Table 3 ijms-22-11240-t003:** Partially methylated alditol acetates generated from the Lcr LPS.

Residue	Area %
Terminal Ribofuranosyl Residue (t-Rib*f*)	3.0
Terminal Rhamnopyranosyl Residue (t-Rha)	0.7
2-Linked Ribofuranosyl Residue (2-Rib*f*)	5.7
3-Linked Ribofuranosyl Residue (3-Rib*f*)	13.8
2-Linked Rhamnopyranosyl Residue (2-Rha)	2.7
3-Linked Rhamnopyranosyl Residue (3-Rha)	1.9
Terminal Glucopyranosyl Residue (t-Glc)	6.7
Terminal Galactopyranosyl Residue (t-Gal)	17.8
4-Linked Glucopyranosyl Residue (4-Glc)	9.6
4-Linked Galactopyranosyl Residue (4-Gal)	9.9
3-Linked Galactopyranosyl Residue (3-Gal)	2.7
Terminal *N*-Acetylglucosamine Residue (t-GlcNAc)	1.4
4-Linked *N*-Acetylglucosamine Residue (4-GlcNAc)	6.5
6-Linked *N*-Acetylglucosamine Residue (6-GlcNAc)	16.1
4,6-Linked *N*-Acetylglucosamine Residue (4,6-GlcNAc)	1.3
3,6-Linked *N*-Acetylglucosamine Residue (3,6-GlcNAc)	0.2

**Table 4 ijms-22-11240-t004:** Composition of the prominent ions in the positive and negative mode spectra as seen in Figure 5.

Ion	Observed Mass *	Calculated Mass *	Composition **
**Positive mode ions**			
[M+Na]^+^	1746.33	1746.39	GlcN_2_16:0(3-OH)_2_14:0(3-OH)_2_28:0(27-OH)_1_
[M+Na]^+^	1774.35	1774.43	GlcN_2_16:0(3-OH)_3_14:0(3-OH)_1_28:0(27-OH)_1_
[MNa+Na]^+^	1848.34	1848.34	P1GlcN_2_16:0(3-OH)_2_14:0 3-OH)_2_28:0(27-OH)_1_
[MNa+Na]^+^	1876.33	1876.36	P1GlcN_2_16:0(3-OH)_3_14:0(3-OH)_1_28:0(27-OH)_1_
**Negative mode ions**			
[M-H]^−^	1774.95	1774.32	P1GlcN_2_16:0(3-OH)_1_14:0(3-OH)_3_28:0(27-OH)_1_
[M-H]^−^	1802.97	1802.36	P1GlcN_2_16:0(3-OH)_2_14:0(3-OH)_2_28:0(27-OH)_1_
[M-H]^−^	1830.99	1830.39	P1GlcN_2_16:0(3-OH)_3_14:0(3-OH)_1_28:0(27-OH)_1_
[M-H]^−^	1926.96	1926.02	PEA1P1GlcN_2_16:0(3-OH)_3_14:0(3-OH)_1_28:0 (27-OH)_1_

Ions marked with an obelisk denote bis-sodiated species. Ions marked with an asterisk denote tri-sodiated species.* Exact mass values. ** Other fatty acyl combinations may be possible, however, the ones given are those that include the major fatty acyl components observed.

**Table 5 ijms-22-11240-t005:** Proton and carbon peak assignments of the delipidated *O*-polysaccharide from Lcr. Carbon chemical shifts in italics.

No.	Residue	H-1	H-2	H-3	H-4	H-5	H-6	H-7	H-8
		*C-1*	*C-2*	*C-3*	*C-4*	*C-5*	*C-6*	*C-7*	*C-8*
**A**	2-β-Rib*f*-1-	5.50	4.22	4.23	3.98	3.67/3.84			
		*109.8*	*84.0*	*72.6*	*85.3*	*65.5*			
**B**	2-β-Rib*f*-1-	5.42	4.20	4.32	3.99	3.66/3.81			
		*109.2*	*83.7*	*73.1*	*85.8*	*65.7*			
**C**	2-β-Rib*f*-1-	5.40	4.21	4.33	4.00	3.66/3.81			
		*109.2*	*83.7*	*72.9*	*85.8*	*65.7*			
**D**	2-β-Rib*f*-1-	5.39	4.21	4.33	4.01	3.66/3.82			
		*109.2*	*83.7*	*73.0*	*85.8*	*65.7*			
**E**	3-β-Rib*f*-1-	5.33	4.33	4.21	4.14	3.71/3.88			
		*111.0*	*77.4*	*81.9*	*83.5*	*65.3*			
**F**	3-β-Rib*f*-1-	5.32	4.34	4.23	4.14	3.71/3.81			
		*111.1*	*77.4*	*82.0*	*83.9*	*65.6*			
**G**	3-β-Rib*f*-1-	5.22	4.32	4.31	4.17	3.70/3.85			
		*110.3*	*77.2*	*82.3*	*83.9*	*65.2*			
**I**	4-α-Glc*p*-1-	5.15	3.59	3.94	3.71	4.24	3.80/3.90		
		*102.3*	*73.9*	*74.3*	*80.6*	*73.1*	*62.2*		
**J**	3-α-Rha*p*-1-	5.05	4.31	4.00	3.62	3.89	1.30		
		*104.7*	*72.7*	*82.7*	*73.8*	*72.4*	*19.3*		
**K**	3-α-Rha*p*-1-	5.02	4.14	3.92	3.54	3.84	1.28		
		*104.9*	*72.6*	*80.9*	*74.0*	*71.9*	*19.3*		
**L**	4-α-Glc*p*NAc-1-	5.01	3.96	3.88	3.68	4.04	3.72/3.79		
		*96.8*	*56.6*	*72.3*	*80.1*	*73.6*	*62.5*		
**M**	3-α-Rha*p*-1-	4.91	4.08	3.76	3.52	4.05	1.26		
		*103.4*	*69.8*	*78.2*	*72.9*	*71.8*	*19.3*		
**N**	4-β-Gal*p*-1-	4.65	3.60	3.78	4.02	3.72			
		*107.1*	*73.8*	*75.5*	*78.4*	*77.9*			
**O**	3-β-Gal*p*-1-	4.51	3.67	3.72	4.00	3.74			
		*105.4*	*73.2*	*82.8*	*71.2*	*78.0*			
**P**	4-β-Gal*p*-1-	4.50	3.57	3.76	4.03	3.73	3.75/3.77		
		*106.3*	*73.3*	*75.5*	*78.5*	*77.2*	*63.7*		
**Q**	4-β-Gal*p*-1-	4.49	3.57	3.77	4.05	3.73	3.75/3.77		
		*106.3*	*73.3*	*75.5*	*78.5*	*77.2*	*63.7*		
**R**	5-α-Kdo*p*			1.86/2.07	4.14	4.09	3.83	4.05	3.62/3.80
		*179.2*	*99.2*	*37.0*	*68.7*	*78.1*	*74.1*	*71.1*	*65.7*
**S**	3-OAc-β-Rib*f*-1-	5.26	4.43	5.11	4.26	3.70/3.81			
		*110.9*	*75.9*	*76.2*	*84.0*	*65.6*			
**T**	2-OAc-β-Rib*f*-1-	5.19	5.19	4.44	4.07	3.70/3.86			
		*110.6*	*79.3*	*72.1*	*85.8*	*65.2*			
**U**	β-Gal*p*-1-	4.64	3.58	3.67	3.92	3.69			
		*107.1*	*74.5*	*75.3*	*71.3*	*77.8*			
**V**	α-Rha*p*-1-	4.98	4.21	3.95	3.48	3.83	1.30		
		*104.8*	*73.2*	*72.6*	*74.8*	*71.8*	*19.3*		
**W**	β-Gal*p*-1-	4.49	3.65	3.72	3.93				
		*106.1*	*73.3*	*75.2*	*71.3*				

**Table 6 ijms-22-11240-t006:** Comparison of measured vs. calculated NMR intensity of anomeric protons based upon the number of residues in proposed structures in Figure 10.

Residue	Calculated NMR Intensity	Measured NMR Intensity
**R**	1.00	0.87
**I**	1.00	1.13
**O**	1.00	0.96
**K**	1.00	1.04
**V**	0.19	0.22
**J**	0.81	0.87
**U**	0.22	0.26
**N**	0.59	0.53
**F**	0.59	0.78
**Q**	2.96	3.48
**E**	2.96	2.17
**W**	0.21	0.25
**C**	3.40	2.70
**G**	3.40	3.57
**P**	3.40	3.48
**A**	3.40	3.22
**B**	3.40	2.43
**S**	0.14	0.17
**T**	0.24	0.28

The molar amount of each structure was estimated by the peak area of the non-reducing end residues S, T, U, V, and W. The measured peak areas were normalized to the average of residues R, I, O, and K, which are present in each structure. The measured peak areas were obtained by performing a line fitting procedure of the anomeric protons in MNova. Residues that did not show well-separated anomeric signals were estimated from the H4 signals in the 2D HSQC spectrum.

## Data Availability

Not applicable.

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
