# Peer review of "Structure of Lipopolysaccharide from Liberibacter crescens Is Low Molecular Weight and Offers Insight into Candidatus Liberibacter Biology"

_ijms, 2021, doi:10.3390/ijms222011240_

Round 1

Reviewer 1 Report

In this manuscript, analytical results of L. crescens LPS are clearly described. The experimental methods used in this study are reliable and the conclusive lipid A structure is properly shown. However, several points need to be improved. The quality of drawing of  polysaccharide (oligosaccharide) structures in Fig. 10 is very poor and the structure cannot be recognized. Probably because of this problem, I could not understand the definition or difference of O-polysaccharide and core-oligosaccharide. These points should be improved to make the interpretation of NMR data easy to understand for the readers. Additionally, the experiment in Fig. 3 and Table 1 needs more explanation. How were the percentages of each sugar calculated from the peaks in Fig. 3? Why are there several peaks for one sugar or one fatty acid? Abbreviations for fatty acids in Fig. 4 are different from those in Fig. 3 or in Text, and they should be unified.

minor points:

  1. Throughout the manuscript, all of bacterial names should be italicized, and with one space after dot like E. coli. The description for all Salmonella strains should be like "Salmonella enterica serovar Minnesota" with strain number.
  2. In Fig. 1, ladder of L. crescens LPS is very faint and cannot be recognized. Can this figure be improved? The abbreviation of the genus name in the figure is "L", not "S".
  3. page 4, line 161: RNAse -> RNase
  4. page 5, line 184: "nor-malizing" -> normalizing
  5. page 6, line 194: C16:0 is saturated fatty acid, not unsaturated one.
  6. page 6, line 219: "spectrums" -> spectra
  7. page 7, line 231: "spectra show" -> spectrum shows
  8. page 7, line 234: Figure 5 -> Figure 6
  9. Fig. 6: Why are the peaks of m/z1746 and 1774 not detected in negative mode (they should be m/z1722 and 1750 in negative mode)?
  10. page 9, line 274: "chromatogram" -> spectrum
  11. page 9, line 283: "peaks" > signals
  12. from page 9 to 17: 1H or 13C should be 1H or 13C.
  13. page 13, line 366: The letter should be  alpha instead of strange spiral mark.
  14. Fig. 11: "Na" in the figure should be removed. Phosphate should be drawn at the C1 position with dotted line, because phosphate is partially present.
  15. page 20, line 548: Brucellae genus -> Brucella spp.
  16. page 21, line 596: add HCl concentration
  17. page 21, line 603: add temperature program of GC
  18. page 22, line 651: NaBD4 -> NaBD4

Author Response

In this manuscript, analytical results of L. crescens LPS are clearly described. The experimental methods used in this study are reliable and the conclusive lipid A structure is properly shown. However, several points need to be improved. The quality of drawing of polysaccharide (oligosaccharide) structures in Fig. 10 is very poor and the structure cannot be recognized. Probably because of this problem, I could not understand the definition or difference of O-polysaccharide and core-oligosaccharide. These points should be improved to make the interpretation of NMR data easy to understand for the readers.

We have replaced Figure 10 with a higher quality version and turned it 90 degrees to make better use of the space on the page. Our hope is that this higher quality figure will allow the reader to clearly visualize the structures we elucidated from the NMR data.

 Additionally, the experiment in Fig. 3 and Table 1 needs more explanation. How were the percentages of each sugar calculated from the peaks in Fig. 3?

We have included a few sentences (on Page 5, line 191) noting that we used standards to make response factors for the carbohydrates and that this allowed us to quantitate them as seen in Table 1.

 Why are there several peaks for one sugar or one fatty acid?

In contrast to the alditol acetates (AA) analysis, the TMS derivatization used for Figure 3 does not include the reduction of anomeric carbons. As a result, each monosaccharide is present as a mixture of α and β anomers and pyranose and furanose forms. Thus, the method can result in up to 4 peaks per monosaccharide.

Abbreviations for fatty acids in Fig. 4 are different from those in Fig. 3 or in Text, and they should be unified.

We have corrected this error and unified the nomenclature for the fatty acids in the text and in Figures 3 and 4. We are grateful to the reviewer for this suggestion.

  1. Throughout the manuscript, all of bacterial names should be italicized, and with one space after dot like E. coli. The description for all Salmonella strains should be like "Salmonella enterica serovar Minnesota" with strain number.

We have italicized all of the bacterial names as suggested. Thank you for the comment.

  1. In Fig. 1, ladder of L. crescens LPS is very faint and cannot be recognized. Can this figure be improved? The abbreviation of the genus name in the figure is "L", not "S".

We have prepared a new Figure 1 that is of a higher quality. We also have added a panel B, which shows lane 6 and 7 of the gel after overexposure to the silver stain. This overexposure made it easier to discern the faint ladder. We have also changed the S to L as suggested.

  1. page 4, line 161: RNAse -> RNase

We have corrected this. Thank you for the suggestion.

  1. page 5, line 184: "nor-malizing" -> normalizing

      We have made this correction. Thank you for the suggestion.

  1. page 6, line 194: C16:0 is saturated fatty acid, not unsaturated one.

We have changed the text to correct this. Thank you for the suggestion.

  1. page 6, line 219: "spectrums" -> spectra

We have corrected this error. Thank you for the suggestion.

  1. page 7, line 231: "spectra show" -> spectrum shows

We have changed the text to correct this. Thank you for the suggestion.

  1. page 7, line 234: Figure 5 -> Figure 6

We have fixed this error. Thank you for the suggestion.

  1. Fig. 6: Why are the peaks of m/z1746 and 1774 not detected in negative mode (they should be m/z1722 and 1750 in negative mode)?

Ions 1746 and 1774 in Figure 6 (top) represent molecular species that do not contain phosphate groups. As such, these ions do not contain negatively charged functionalities and are therefore not present in the negative ion mode MALDI spectrum (Figure 6, bottom).

  1. page 9, line 274: "chromatogram" -> spectrum

We have changed the text to correct this. Thank you for the suggestion.

  1. page 9, line 283: "peaks" > signals

 We have changed the text to read signals instead of peaks. Thank you for the suggestion.

  1. from page 9 to 17: 1H or 13C should be 1H or 13C.

We have gone through the text and changed all 1H and 13C to 1H and 13C. Thank you for catching the mistake.

  1. page 13, line 366: The letter should be alpha instead of strange spiral mark.

We have changed the text to correct this. Thank you for the suggestion.

  1. Fig. 11: "Na" in the figure should be removed. Phosphate should be drawn at the C1 position with dotted line, because phosphate is partially present.

We have re-drawn Figure 11 with a dotted line connecting a phosphate to the proximal glucosamine. We have also removed the sodium symbol. Thank you for the suggestion.

  1. page 20, line 548: Brucellae genus -> Brucella spp.

We have fixed this error. Thank you for the suggestion.

  1. page 21, line 596: add HCl concentration

We have added the molar concentration of methanol HCl to the method section. Thank you for the suggestion.

  1. page 21, line 603: add temperature program of GC

We have added the GC temperature program to the method section. Thank you for the suggestion.

  1. page 22, line 651: NaBD4 -> NaBD4

We have fixed this error. Thank you for the suggestion.

Reviewer 2 Report

Huanglongbing (HLB) disease plagues the citrus industry and leads to billions of dollars in crop losses. Thus it is important to study the causative agent of the disease, Candidatus Liberibacter asiaticus. A closely related culturable bacterium (Liberibacter crescens or Lcr) has been studied to improve understanding about the Liberibacter genus. In this study Black et al. determined the structure of lipopolysaccharide of Lcr, which is a major component of the bacterial outer membrane and plays an important role in bacterial virulence. Using a collection of techniques including NMR and LC-MS, the authors were able to determine the molecular structure of LPS. The over study is solid, the data presented fully support the conclusion, and the determination of the LPS structure is an important discovery on the biochemistry of the Lcr, and the Liberibaceter genus. 

Author Response

We would like to thank reviewer 2 for their comments and express our appreciation for their time and effort.